# Oral Formulation Based on Irbesartan Nanocrystals Improve Drug Solubility, Absorbability, and Efficacy

**DOI:** 10.3390/pharmaceutics14020387

**Published:** 2022-02-10

**Authors:** Noriaki Nagai, Fumihiko Ogata, Ayari Ike, Yurisa Shimomae, Hanano Osako, Yosuke Nakazawa, Naoki Yamamoto, Naohito Kawasaki

**Affiliations:** 1Faculty of Pharmacy, Kindai University, 3-4-1 Kowakae, Higashiosaka 577-8502, Osaka, Japan; ogata@phar.kindai.ac.jp (F.O.); 1711610031w@kindai.ac.jp (A.I.); 1811610085e@kindai.ac.jp (Y.S.); 1911610054p@kindai.ac.jp (H.O.); kawasaki@phar.kindai.ac.jp (N.K.); 2Faculty of Pharmacy, Keio University, 1-5-30 Shibakoen, Minato 105-8512, Tokyo, Japan; nakaza-wa-ys@pha.keio.ac.jp; 3Research Promotion and Support Headquarters, Center for Clinical Trial and Research Support, Fujita Health University, 1-98 Dengakugakubo, Kutsukake, Toyoake 470-1192, Aichi, Japan; naokiy@fujita-hu.ac.jp

**Keywords:** irbesartan, nanocrystalline, tablet, intestinal absorption, endocytosis

## Abstract

We previously reported that the bioavailability (BA) of irbesartan (IRB), a BSC class II drug, was improved by preparing nanocrystalline suspensions. However, nanocrystalline suspensions have chemical and physical instabilities and must be converted into tablets through drying approaches in order to overcome such instabilities. In this study, we attempted to design a molded tablet based on nanocrystalline IRB suspensions (IRB-NP tablet) and investigated the effects of this IRB-NP tablet on blood pressure (BP) in a stroke-prone spontaneously hypertensive (SHR-SP) rat. The IRB-NP tablet (with a hardness of 42.6 N) was developed by combining various additives (methylcellulose, 2-hydroxypropyl-β-cyclodextrin HPβCD, D-mannitol, polyvinylpyrrolidone, and gum arabic) followed by bead-milling and freeze-drying treatments. The mean particle size in the redispersions of the IRB-NP tablet was approximately 118 nm. The solubility and intestinal absorption of IRB in the IRB-NP tablet were significantly enhanced in comparison with the microcrystalline IRB tablet (IRB-MP tablet), and both solubility and clathrin-dependent endocytosis helped improve the low BA of the IRB. In addition, the BP-reducing effect of the IRB-NP tablet was significantly higher than that of the IRB-MP tablet. These results provide useful information for the preservation of nanocrystalline suspensions of BCS class II drugs, such as IRB.

## 1. Introduction

Hypertension therapy reduces the risk for the onset of stroke and cardiovascular disease, which often coexist [1,2]. Presently, angiotensin-converting enzyme (ACE) inhibitors and angiotensin II receptor blockers (ARBs) are used worldwide to treat hypertension; the antihypertensive effects of ACE inhibitors and ARBs are led by the inhibition of angiotensin II production [3]. On the other hand, the inhibition of the ACE also causes the production of bradykinin and substance P. Therefore, angioedema and dry cough were observed as side effects among patients undergoing ACE-inhibitor treatment. In contrast, ARBs provide a significant blood pressure (BP)-reducing effect without these side effects, since ARBs inhibit the interaction between the angiotensin II type-1 receptor and angiotensin II [4,5]. Thus, ARBs are useful for hypertension therapy.

Irbesartan (IRB), which is classified as a Biopharmaceutic Classification System (BCS) class II drug (low solubility and high permeability), is an ARB that is primarily used for the treatment of cardiac insufficiency, cardiac arrhythmia, and hypertension [6,7]. The pKa of IRB is 4.12 and 7.4, and the molecular weight is 428.53 g/mol [8]. Two crystalline forms of IRB (form A and B) have been reported so far. The solubility of form A and B under room temperature is approximately 0.011 and 0.0006 mg/mL, respectively [9,10]. The *C*_max_ values (peak plasma concentrations) were recorded to be within 1.5–2 h after the oral administration of IRB [11], and the average absolute bioavailability (BA) of IRB oral administration was found to be approximately 60–80% [12]. On the other hand, low solubility limits the BA of IRB. Moreover, IRB exhibits a high variation in oral BA between individuals [13].

To mitigate the issues associated with BCS class II drugs, many techniques have been reported, including the use of nanocrystalline suspensions, micelles, amorphous molecules, nanoparticles, liposomes, cyclodextrin complexes, co-solvents, and salts [14,15,16,17]. Nanocrystalline suspensions also improve the oral BA of BCS class II drugs by enhancing the drug dissolution rates, solubility, and intestinal penetration due to a high surface area and cell uptake via energy-dependent endocytosis [18,19,20]. In addition, we previously found that the intestinal absorption of IRB was enhanced by preparing nanocrystalline suspensions via bead-milling treatment and that both high solubility and clathrin-dependent endocytosis (CME) are related to the improvement of low BA [21]. However, nanocrystalline suspensions are thermodynamically unstable since they possess high Gibbs free energy due to their very large exposed surface areas [21] and are thus attacked by microbes. Consequently, IRB is prone to chemical and physical instabilities, such as hydrolysis, agglomeration, and Ostwald ripening [22]. In order to overcome these issues, it is important to convert the nanocrystalline suspensions to dried powders, tablets, or capsule formulations for oral administration [23,24,25,26]. In this study, we attempted to prepare suspensions containing nanocrystalline IRB (IRB-NP suspensions) via the break-down method (bead milling). We designed the molded tablets based on nanocrystalline IRB (IRB-NP tablet). In addition, we investigated the effect of the IRB-NP tablet on BP in stroke-prone spontaneously hypertensive (SHR-SP) rats [27,28].

## 2. Materials and Methods

### 2.1. Chemicals and Animals

All other chemicals were of the highest purity commercially available. Briefly, 2-hydroxypropyl-β-cyclodextrin (HPβCD, average molar substitution 0.6, average molecular weight 1380) was purchased from Nihon Shokuhin Kako Co., Ltd. (Tokyo, Japan). A Bio-Rad Protein Assay Kit was provided by BIO-RAD (Hercules, CA, USA), and rottlerin and dynasore were provided by Nacalai Tesque (Kyoto, Japan). IRB powder (IRB microparticles, IRB-MP), D-mannitol, cytochalasin D, gum arabic, polyvinylpyrrolidone (PVP, average molecular weight 40,000), and propyl p-hydroxybenzoate were obtained from Wako Pure Chemical Industries, Ltd. (Osaka, Japan). Nystatin was provided by Sigma-Aldrich Japan (Tokyo, Japan). Methylcellulose (MC, average molecular weight 25,000) and pentobarbital were purchased from Tokyo Chemical Industry Co., Ltd. (Tokyo, Japan). In addition, all experiments on animals were performed according to the guidelines of Kindai University and the Japanese Pharmacological Society. Fourteen-week-old male SHR-SP rats (approximately 340 g) and 6-week-old male Wistar rats (approximately 200 g) were provided from the Faculty of Medicine of Kindai University and Kiwa Laboratory Animals Co., Ltd. (Wakayama, Japan), respectively. The rats were housed at 25 °C under normal conditions. Water and a standard diet CE-2 (Clea Japan Inc., Tokyo, Japan) were provided freely. The experiments were approved on 1 April 2019 by Kindai University under project code KAPS-31-014.

### 2.2. Preparation of the IRB-NP Tablet

Suspensions containing IRB microparticles were prepared by mixing IRB microparticles, MC, and HPβCD in purified water (IRB-MP suspensions). After that, D-mannitol, PVP, and gum arabic were added into the IRB-MP suspensions, and the 700 µL of IRB suspension was transferred to press-through-pack (PTP) sheet. Then, the suspensions were frozen for 24 h at −80 °C, and freeze-dried by FREEZE DRYER FD-1000 (TOKYO RIKAKIKAI Co., Ltd., Tokyo, Japan) for 2 days (IRB-MP tablet). On the other hand, the suspensions containing IRB nanoparticles were prepared following our previous reports using Bead Smash 12 (a bead mill, Wakenyaku Co. Ltd., Kyoto, Japan) [20,29]. Then, 0.1 mm zirconia beads were added into IRB-MP suspensions and milled using a Bead Smash 12 at 5500 rpm for 30 s × 30 times at 4 °C (IRB-NP suspensions). Afterward, the D-mannitol, PVP, and/or gum arabic were added into the milled IRB suspensions, and the 700 µL of IRB suspension was transferred to PTP sheet. After that, the suspensions were chilled for 24 h at −80 °C, freeze-dried by the FREEZE DRYER FD-1000 for 2 days, and used as the IRB-NP tablets in this study. The compositions of the IRB suspensions used to prepare the molded tablets in this study are shown in Table 1. The IRB tablet was stored under 22 °C conditions in the stability test. The tablet form (Rp.8) was as follows: major axis 11.90 ± 0.01 mm; minor axis 7.11 ± 0.01 mm; thickness 6.82 ± 0.01 mm; weight 150.7 ± 1.1 mg; and mean ± S.E., *n* = 30. The shape and dimensions of the tablets were analyzed by the image analyzing software, Image J. To evaluate the redispersion and intestinal absorption, the IRB tablets were suspended in purified water and used.

### 2.3. Characterization of the IRB-NP Tablets and It’s Redispersion

An LC-20AT HPLC system was used to measure the IRB concentration (Shimadzu Corp., Kyoto, Japan), and 1 µg/mL propyl p-hydroxybenzoate was selected as an internal standard. The column, wavelength for detection, and mobile phase were as follows: 2.1 × 50 mm Inertsil^®^ ODS-3 column at 35 °C (GL Science Co., Inc., Tokyo, Japan), 254 nm, and 0.1% formic acid/acetonitrile (63/37, *v*/*v*%), respectively. The mobile phase flowed at 0.25 mL/min. The solubility of IRB in the suspensions and tablet was evaluated as follows: The IRB in the suspensions and redispersions of the IRB tablet were separated into non-solubilized and soluble IRB via centrifugation at 100,000× *g* using a Beckman Optima^TM^ MAX-XP Ultracentrifuge (Beckman coulter, Osaka, Japan). Then, the soluble IRB was collected, and the levels were measured using the HPLC method described above. The particle size was measured using a laser diffraction particle-size analyzer SALD-7100 (Shimadzu Corp. Kyoto, Japan) and a dynamic light scattering NANOSIGHT LM10 (QuantumDesign Japan, Tokyo, Japan). Atomic force microscopic (AFM) images of the IRB-NP were taken using an SPM-9700 (Shimadzu Corp., Kyoto, Japan), and a micro-electrophoresis zeta potential analyzer, model 502, was used to measure the zeta potential (Nihon Rufuto Co., Ltd., Tokyo, Japan). The crystalline form of the IRB was measured in the suspensions and tablet using a powder X-ray diffraction (XRD) analyzer, Mini Flex II (Rigaku Co., Tokyo, Japan). In addition, the melting point was analyzed via thermogravimetry–differential thermal analysis (TG-DTA) measurements using a DTG-60H (Shimadzu Corp., Kyoto, Japan). The viscosity at 22 °C was determined using an SV-1A (A & D Company, Limited, Tokyo, Japan). The dispersibility of the IRB suspensions and the redispersions of the IRB tablet were observed as follows: Three milliliters of the IRB suspensions and redispersions of the IRB tablet were incubated in a 5 mL tube under dark conditions at 22 °C for one month. The samples were collected from the upper 90% of the test tube over time. In this study, we measured changes of IRB levels in the samples to evaluate the dispersibility. The dissolution, disintegration, and friability of IRB tablet were analyzed according to the 18th edition of the *Japanese Pharmacopoeia* (JP), and the testers for measuring dissolution, disintegration, and friability were used—NTR-1000 (TOYAMA SANGO Co. Ltd., Osaka, Japan), NT-2H (TOYAMA SANGO Co. Ltd., Osaka, Japan), and oriental reaction motor (ORIENTAL MOTOR Co., Ltd., Tokyo, Japan), respectively. These characteristics were analyzed using the methods in our previous study [19,20,29,30].

### 2.4. In Vitro Intestinal Penetration in the IRB Table

A methacrylate cell was used to measure the intestinal penetration of IRB in the tablet according to a previous study [19]. The jejunums were collected from 7-week-old Wistar rats and set on the methacrylate cell. The reservoir (3 mL) was filled with a pH 7.4 buffer (basolateral side), and the donor (apical) side (3 mL) was filled with redispersions of the IRB-NP tablet (400 μm). The permeation area and dimensions of the intestinal tissue are 0.3485 cm^2^ and 0.1 cm (average of 5 rats), respectively. These samples were collected from the reservoir side over time, and the IRB levels in the sample were measured via the HPLC method. Moreover, the area under the drug concentration–time curve in the reservoir chamber (*AUC*_0–6 h_) was calculated based on the permeation behavior by using the trapezoidal rule. In this study, the function of energy-dependent endocytosis in the small intestine was universally inhibited via incubation under cold conditions (4 °C) [31]. In addition, each energy-dependent endocytosis (caveolae-dependent endocytosis (CavME), clathrin-dependent endocytosis (CME), macropinocytosis (MP), and phagocytosis) process was prevented via treatment with nystatin (54 μm) [32], dynasore (40 μm) [33], rottlerin (2 μm) [34], and cytochalasin D (10 μm) [32], respectively. These pharmacological inhibitors were prepared via 0.5% dimethyl sulfoxide (DMSO, vehicle), and treatment of pharmacological inhibitors was performed 5 min before the start of the experiment until the end of the experiment.

### 2.5. Measurement of the Plasma IRB Concentration

A cannula was inserted into the right jugular vein of the 7-week-old Wistar rats 1 day before the experiment to collect blood, and the rats were fasted for 8 h. Afterward, redispersions of the IRB-MP or IRB-NP tablet (0.2 mg/kg) were orally administered, and venous blood (200 μL) was collected through the cannula over time. The collected blood was then centrifuged at 800× *g* for 15 min at 4 °C, and the IRB concentrations in the supernatant (plasma IRB) were measured via HPLC, as described above. The trapezoidal rule was used to calculate the area under the blood concentration–time curve (*AUC*_0–24 h_).

### 2.6. Measurement of Blood Pressure (BP)

Redispersions of the IRB-MP or IRB-NP tablets (0.2 mg/kg) were orally administered to 14-week-old SHR-SP rats, and the heart rate, systolic (SBP), and diastolic (DPB) blood pressures were measured using a noninvasive blood pressure analysis system, BP-98A (Softron, Tokyo, Japan). The ΔSBP and ΔDBP were used to calculate the difference in the BP between rats administered with and without the IRB tablet, and the trapezoidal rule was used to estimate the area under the ΔBP (ΔSBP and ΔDBP)–time curve for 0–24 h (*AUC*_ΔSBP_ and *AUC*_ΔDBP_).

### 2.7. Statistical Analysis

The data are expressed as the mean ± standard error (S.E.). A statistical analysis was performed using Student’s *t*-test and ANOVA followed by Dunnett’s multiple comparison, with *p* < 0.05 considered to be significant.

## 3. Results

### 3.1. Design of the IRB-NP Tablet and Evaluation of Their Characteristics

We previously reported that suspensions containing nanocrystalline IRB can be produced via combinations of various additives and the break-down method [19]. Following the previous study, we first prepared suspensions containing IRB nanoparticles. The particle sizes of IRB in the mixture, consisting of IRB-MP, MC, and HPβCD, were reduced to 50–202 nm (mean particle size 87 ± 11 nm) from 0.51 to 48 µm (mean particle size 4.89 ± 0.43 µm) via bead-mill treatment, with a sphere-shaped form (Figure 1). Next, we designed a tablet containing IRB nanoparticles based on the IRB-NP suspensions. Figure 2A presents an image of the IRB tablet after the IRB-NP suspensions were freeze-dried, and Figure 2B and Table 2 present the microscopic images (Figure 2B) and characteristics (Table 2) for the redispersions of the IRB tablets. The freeze-dried IRB-NP with D-mannitol (Rp.3) easily disintegrated and could not maintain their shapes as tablets.

On the other hand, the addition of PVP (Rp.4) and gum arabic (Rp.5) maintained the tablets’ shapes. However, the IRB particles in the redispersions of the Rp.4 and Rp.5 formulations were micro-sized. In contrast to the results of the Rp.3–5 formulations, the problem was improved using a combination of D-mannitol, PVP, and gum arabic (Rp.7). The Rp.7 formulation maintained its shape as a tablet, and the IRB particles in the redispersions of the Rp.7 formulation were nanoparticles.

We investigated the solubility, viscosity, and zeta potentials in the redispersions of IRB tablets (Table 2). The solubility of IRB was similar between the IRB-MP suspension (Rp.1) and IRB-MP tablet. On the other hand, the solubility of the IRB-NP suspensions (Rp.2) was higher than that of the IRB-MP suspension (Rp.1), and the combination of additives (D-mannitol, PVP, and gum arabic) and bead-mill treatment enhanced the solubility of IRB. The viscosity was increased by the gum arabic content in the IRB tablets.

In contrast to the results for viscosity, the zeta potentials decreased with gum arabic content in the IRB tablets. Furthermore, we observed the effects of gum arabic content on the hardness, friability (Figure 3), and dispersibility of the IRB redispersions of the IRB-NP tablet (Figure 4). An increase in gum arabic contents enhanced the hardness and decreased the friability of the IRB-NP tablets. The particle size of IRB remained nano-ordered in the redispersions of the IRB-NP tablets, and the IRB particle size in the redispersions of 1-month-stored IRB-NP tablets was also approximately 60–350 nm (Table 3). On the other hand, the gum arabic contents decreased the dispersibility of IRB in the redispersed IRB-NP tablet, and a significant aggregation of IRB was observed in the Rp.9 and Rp.10 formulations containing 12 and 16 *w*/*w*% gum arabic (Figure 4). Figure 5 shows the powder X-ray diffraction patterns and TG-DTA curve of the Rp.8 formulation. Although these patterns appeared in several superimposed reflections, we selected the peaks at 12 and 17 to identify crystal forms of IRB, since these peaks could be confirmed even after freeze-drying. The XRD patterns in the Rp.8 formulation were similar to those of the IRB-MP tablet, and these peaks at 12 and 17 were observed for both the IRB-MP tablet and the Rp.8 formulation (Figure 5A,B). In the TG-DTA curve, the melting point of IRB shifted from 186 to 165 °C via the addition of additives. On the other hand, the melting point of IRB was similar between the IRB-MP tablet and Rp.8 formulation (Figure 5C,D). In addition, we measured the dissolution, disintegration, and friability of the Rp.8 formulation according to the 18th edition of the JP. The dissolution ratio and disintegration time of the Rp.8 formulation were 99.9 ± 0.4%, 6 ± 1 s, respectively (*n* = 30).

### 3.2. Effect of Energy-Dependent Endocytosis on the Transintestinal Penetration of the IRB-NP Tablet (Rp.8)

It was previously reported that energy-dependent endocytosis is related to the high intestinal absorption of drug nanoparticles [19,20,29]. In this study, we demonstrated the relationships between energy-dependent endocytosis and the transintestinal penetration of an IRB-NP tablet (Rp.8 formulations shown as Table 1) using an in vitro study (Figure 6). The IRB penetrated through the small intestine at 37 °C, and the *AUC*_0–6 h_ values in the IRB-NP tablet (Rp.8) was significantly higher than that in the IRB-MP tablet (*AUC*_0–6 h_, 36 ± 9 nmol∙h/cm^2^). On the other hand, in the IRB-NP tablet (Rp.8), the IRB was detected in the reservoir chamber only in a liquid state. The transintestinal penetration of the IRB-NP tablet (Rp.8) was significantly attenuated under 4 °C conditions, in which energy-dependent endocytosis was inhibited [31]. Furthermore, the *AUC*_0–6 h_ values in the small intestine co-treated with nystatin, dynasore, rottlerin, and cytochalasin D were 136.9, 88.7, 150.3, and 152.1 nmol∙h/cm^2^, respectively. Moreover, the dynasore more significantly attenuated the transintestinal penetration of the IRB-NP tablet (Rp.8) compared to the vehicle group. Figure 7 shows the changes in the plasma IRB levels in the rats orally administered the IRB-NP tablets (Rp.8). The intestinal absorption of IRB in the IRB-NP tablet (Rp.8) was 1.94-fold that of the IRB-MP tablet. These results showed that the transintestinal penetration of the IRB-NP tablet (Rp.8) was promoted by energy-dependent endocytosis, such as CME.

### 3.3. BP-Reducing Effect of the IRB-NP Tablet (Rp.8) in the SHR-SP Rats

It was also important to evaluate the BP-reducing effect of the IRB-NP tablet. Figure 8 shows the ΔSBP and ΔDBP profiles in the SHR-SP rats orally administered the IRB-NP tablet (Rp.8). The SBP and DPB in 14-week-old SHR-SP rats were 268 ± 9.7 mmHg and 146 ± 5.1 mmHg, respectively. Moreover, the IRB-NP tablet (Rp.8) decreased the BP in the SHR-SP rats. The peak BP-reducing effect in SHR-SP rats orally administered the IRB-NP tablet (Rp.8) was observed 2–6 h after oral administration, and the *t*_max_ was shortened in comparison with that observed under the IRB-MP tablet. Moreover, the *AUC*_ΔSBP_ and *AUC*_ΔDBP_ of the IRB-NP tablet (Rp.8) were both significantly higher than those of the IRB-MP tablet, and the BP-reducing effects were similar to the results of the absorption profile (Figure 7).

## 4. Discussion

The application of nanocrystalline suspensions represents a useful strategy to enhance the oral BA of poorly soluble drugs [35,36,37,38]. We also reported that the BA of IRB, a BSC class II drug, was improved by preparing nanocrystalline suspensions [19]. However, nanocrystalline suspensions have chemical and physical instabilities and must be converted into tablets through drying approaches in order to overcome such instabilities. In this study, we used IRB as a candidate drug model for nanocrystalline suspensions and designed an IRB-NP tablet using a combination of bead-milling and freeze-drying techniques. Moreover, we found that the particles of IRB remained nano-ordered when the IRB-NP tablet was redispersed, and showed high solubility, intestinal absorption, and antihypertensive effects compared to traditional IRB powder.

The break-down and build-up methods are both used to produce nanocrystalline suspensions [24]. In this study, we prepared the nanocrystalline suspensions of IRB (IRB-NP suspensions) via the break-down approach according to our previous study [19]. In short, HPβCD and MC were used to prevent the aggregation of IRB particles and increase the crushing efficiency in the bead mill, respectively. The particle sizes of the IRB-NP suspensions under bead-mill treatment were approximately 50–200 nm (Figure 1), and the drug solubility of IRB was significantly higher than that in IRB without bead-mill treatment (Table 2). Next, we attempted to prepare an IRB-NP tablet based on IRB-NP suspensions by using a freeze-drying treatment. Excipients were necessary to produce the tablet, since the tablet shape was not able to be maintained without excipients. A binder, nanoparticle reducing agent, and dispersing agent were also required to produce redispersible tablets based on nanocrystalline suspensions. D-mannitol is widely used in tablet formulations due to its sweet, cool taste and compatibility with a wide range of drugs [39]. Gum arabic is produced from the dried exudates of acacia senegal and acacia seyal [40] and works as a binder for the production of tablets. In addition, PVP is often used in pharmacology as a nanoparticle reducing agent and dispersing agent, growth-affecting agent, and nanoparticle surface stabilizer [41]. These reagents easily dissolve in water and can be applied to the production of tablets. Therefore, we selected D-mannitol, PVP, and gum arabic to produce the tablet containing IRB nanoparticles in this study. The addition of gum arabic enhanced the hardness and maintained the shape of the tablet (Figure 2 and Figure 3). In addition, the IRB particles in redispersions of the Rp.7 formulation with D-mannitol, PVP, and gum arabic were nanoparticles (Table 3 and Figure 2). These results suggest that PVP enhanced the redispersibility of the tablet containing D-mannitol and gum arabic, since nanoparticles were not detected in the redispersions of the Rp.6 formulation without PVP.

We also characterized the IRB-NP tablet. The combination of additives (D-mannitol, PVP, and gum arabic) and bead-mill treatment enhanced the solubility of IRB, while the viscosity was increased with gum arabic content in the IRB tablets. On the other hand, the gum arabic contents decreased the dispersibility of IRB in the redispersed IRB-NP tablets, and the addition of more than 12 *w*/*w*% gum arabic resulted in a significant aggregation of IRB in the redispersions (Figure 4). Based on these results, we used the Rp.8 formulation as the IRB-NP tablet in subsequent experiments. We also determined the powder X-ray diffraction patterns and the TG-DTA curve of the Rp.8 formulation (Figure 5). These operations, using bead-mill and freeze-drying treatments, did not change the crystal structure of IRB, since the powder X-ray diffraction patterns and the TG-DTA curve were similar between the IRB-MP tablet and Rp.8 formulation. In this study, we showed the stability of 1-month-stored IRB-NP tablets (Table 3); however, it is not enough to characterize the ability of tablets to stabilize the nanosuspensions. Therefore, we also monitored the changes in the main properties and microstructures of tables. The redispersion capacity (Appendix A), disaggregation time (6 ± 1 s, *n* = 5), friability (5.1 ± 0.5%, *n* = 5), hardness (42 ± 6 N, *n* = 5), and XRD pattern (Appendix A) in a 3-month-stored Rp.8 formulation were similar to the redispersions of the 1-month-stored Rp.8 formulation. Thus, the Rp.8 formulation was stable for 3 months. On the other hand, relative humidity (RH)%, temperature stability tests, and long-term storage tests are needed for further investigation.

We also investigated the intestinal absorption of IRB in the oral administration of the IRB-NP tablet. The intestinal absorption of IRB in the IRB-NP tablet (Rp.8) was 1.94-fold that in the IRB-MP tablet (Figure 7). Our previous study showed that the intestinal absorption of IRB was enhanced by preparing nanocrystalline suspensions [19] and that both high solubility and clathrin-dependent endocytosis (CME) were related to the improvement of a low BA. Therefore, we measured the effect of energy-dependent endocytosis on the intestinal absorption of IRB via the methacrylate cell set on the removed rat gut (Figure 6). The IRB in the IRB-NP tablet was transferred to the reservoir side (basolateral side) from the donor side (apical side), and liquid IRB was only detected in the basolateral side. This result indicates that the IRB nanoparticles in the IRB-NP tablet dissolved when they permeated the intestine. On the other hand, the transintestinal penetration of the IRB-NP tablet was significantly attenuated at 4 °C (Figure 6A,B) in this study. This result suggests that energy-dependent endocytosis was also related to the high intestinal penetration of the IRB-NP tablet, since all energy-dependent endocytosis was inhibited under low-temperature conditions (4 °C) [31]. Moreover, nystatin (CavME inhibitor), dynasore (CME inhibitor), rottlerin (MP inhibitor), and cytochalasin D (phagocytosis inhibitor) were used to evaluate the relationships between the endocytosis pathway and the transintestinal penetration of the IRB-NP tablet. Treatment with dynasore significantly attenuated the transintestinal penetration of the IRB-NP tablet compared to the vehicle group (Figure 6C,D). From these results, we hypothesized that the IRB nanoparticles in the IRB-NP tablet were taken up into the intestine by the CME pathway, dissolved in the intestinal tissue, and then released into the blood as liquid IRB. Thus, the absorption mechanism of the IRB-NP tablet in this study supported the previous study [19], since the pathway in the IRB-NP tablet was similar to that in nanocrystalline suspensions of IRB.

It was important to select appropriate model animals for evaluating the antihypertensive effects of IRB-NP tablets. SHR and SHR-SP rats are well-known as model animals for hypertension, and SHR-SP is a sub-strain of SHR. Moreover, the BP levels in the SHR-SP rats were significantly enhanced, compared to those in SHR rats, with aging [27,28]. In the normal Wistar rats, DBP and SBP were shown to stabilize at 90–110 and 140–150 mmHg, respectively [42]. However, the DBP and SBP in the SPR-SP rats reached approximately 150 and 250 mmHg, respectively [19,27,28]. Based on this background, we used this animal model (SHR-SP rats) to evaluate the antihypertensive effects under the oral administration of the IRB-NP tablet. The *AUC*_ΔSBP_ and *AUC*_ΔDBP_ of the IRB-NP tablet were both significantly higher than those of the IRB-MP tablet, and the BP-reducing effect was similar to the results of the absorption profile (Figure 7 and Figure 8). In addition, all these mechanisms of intestinal penetration, absorption profiles, and BP-reducing effects were similar to those found in data for nanocrystalline suspensions that we previously reported [19]. These results showed that an oral formulation (IRB-NP tablet) using a combination of various additives (MC, HPβCD, D-mannitol, PVP, and gum arabic) under bead-milling and freeze-drying treatment can produce nanocrystalline suspensions of IRB.

Further studies are needed to investigate how energy-dependent endocytosis is induced in the small intestinal when treated with IRB nanoparticles. Moreover, it is important to clarify whether this technique, using a combination of various additives (MC, HPβCD, D-mannitol, PVP, and gum arabic), bead milling, and freeze-drying, can improve the low BA of other BCS drugs of different classes (class III and IV). In future work, we plan to design a tablet based on drug nanoparticles using BCS class III and IV drugs.

## 5. Conclusions

We designed an IRB-NP tablet using a combination of various additives (MC, HPβCD, D-mannitol, PVP, and gum arabic) along with bead-milling and freeze-drying treatments, and found that the particles of IRB remained nano-ordered when the IRB-NP tablet was redispersed. Moreover, the IRB-NP tablet provided higher solubility and intestinal penetration than the traditional IRB powder. The improvement in intestinal absorption under the oral administration of the IRB-NP tablet was caused by the enhanced solubility and CME (Figure 9). These results provide useful information for the preservation of nanocrystalline suspensions of BCS class II drugs, such as IRB.

## Figures and Tables

**Figure 1 pharmaceutics-14-00387-f001:**
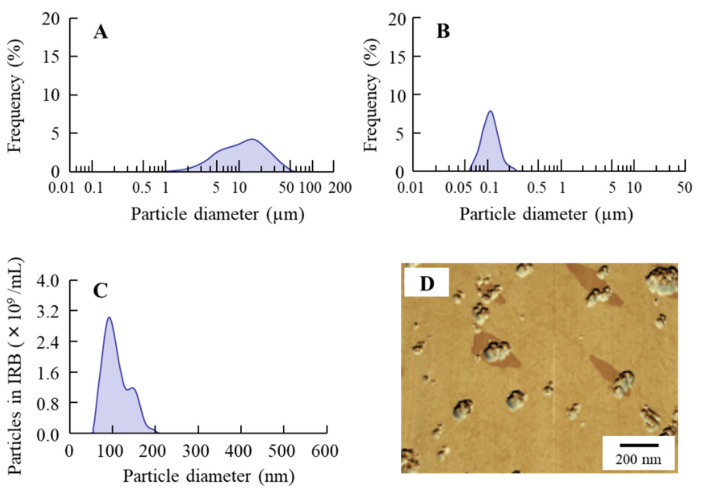
Particle sizes of IRB in IRB-NP suspensions. (**A**) Particle size frequencies of IRB in IRB-MP suspensions as determined by the SALD-7100; (**B**,**C**) particle size frequencies of IRB in IRB-NP suspensions determined by the (**B**) SALD-7100 and (**C**) NANOSIGHT LM10, respectively; and (**D**) AFM image of the IRB in IRB-NP suspensions. Bars show 200 nm. The particle size of the IRB-NP suspensions was approximately 50–200 nm.

**Figure 2 pharmaceutics-14-00387-f002:**
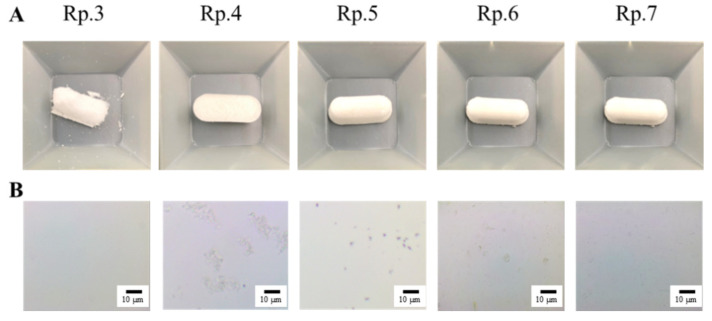
Image of the IRB-NP tablets shown in Table 1. (**A**) Photograph of IRB tablets (Rp. 3–7); (**B**) microscopic image in redispersed IRB tablet (Rp.3–7). IRB tablets were suspended in purified water. Bars show 10 µm. The form of the tablet based on IRB-NP suspensions was maintained by the addition of D-mannitol, PVP, and gum arabic (Rp.7), and the IRB particles in the redispersion were nanoparticles.

**Figure 3 pharmaceutics-14-00387-f003:**
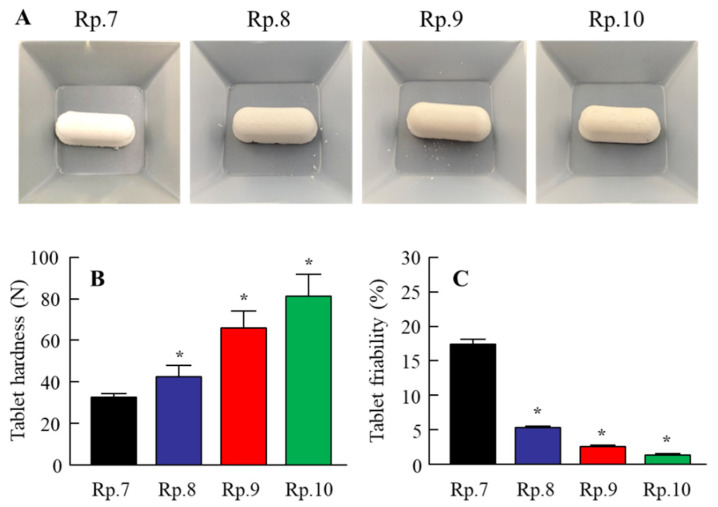
Effect of gum arabic on the hardness and friability of the IRB-NP tablets shown in Table 1. (**A**–**C**) Photograph (**A**), hardness (**B**), and friability (**C**) of the IRB-NP tablets (Rp.7–10). *n* = 10. * *p* < 0.05 vs. Rp.7 for each category. The hardness of the IRB-NP tablets increased with gum arabic contents. Moreover, the addition of gum arabic decreased the friability of the IRB-NP tablets.

**Figure 4 pharmaceutics-14-00387-f004:**
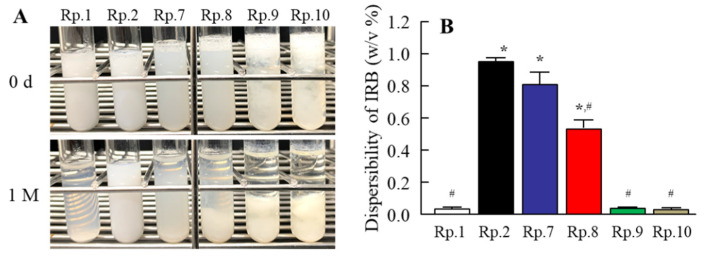
(**A**) Images and (**B**) dispersibility of the IRB immediately (0 d) and 1 m after redispersion of the IRB-NP tablet. The formulation of each Rp. is shown in Table 1, and the IRB tablets were suspended by purified water. *n* = 6. * *p* < 0.05 vs. Rp.1 for each category. # *p* < 0.05 vs. Rp.2 for each category. The gum arabic contents decreased the dispersibility of IRB in the redispersed IRB-NP tablets. In particular, the addition of more than 12 *w*/*w*% gum arabic resulted in a significant aggregation of IRB in the redispersions.

**Figure 5 pharmaceutics-14-00387-f005:**
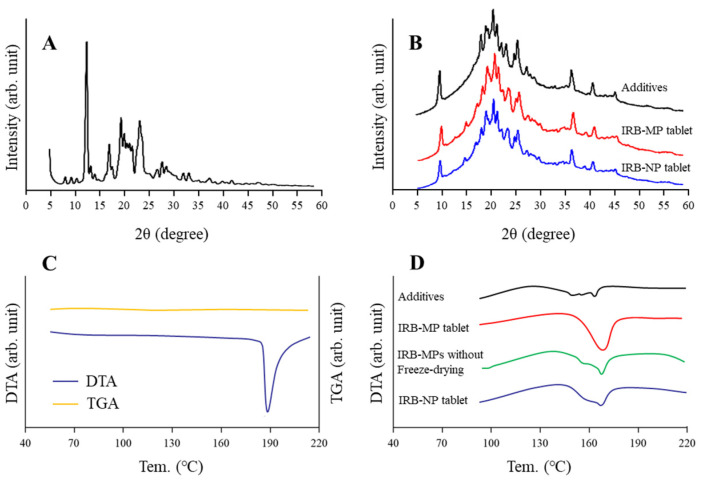
Analysis of the crystalline structure of the IRB-NP tablet (Rp.8) shown in Table 1. (**A**) The powder X-ray diffraction patterns of the IRB reagent without additives and bead-mill treatment; (**B**) the powder X-ray diffraction patterns for the additives (vehicle for the IRB tablet), IRB-MP tablet, and IRB-NP tablet; (**C**) TG-DTA curves of the IRB reagent without additives and bead-mill treatment; (**D**) the DTA curves of additives (vehicle for IRB tablet), IRB-MP tablet with or without freeze-drying, and IRB-NP tablet. The peaks used to identify crystal forms of IRB were detected at 12 and 17 in the XRD, and the peaks were confirmed in the IRB-NP tablet. In addition, the melting point of IRB was similar between the IRB-MP and IRB-NP tablets.

**Figure 6 pharmaceutics-14-00387-f006:**
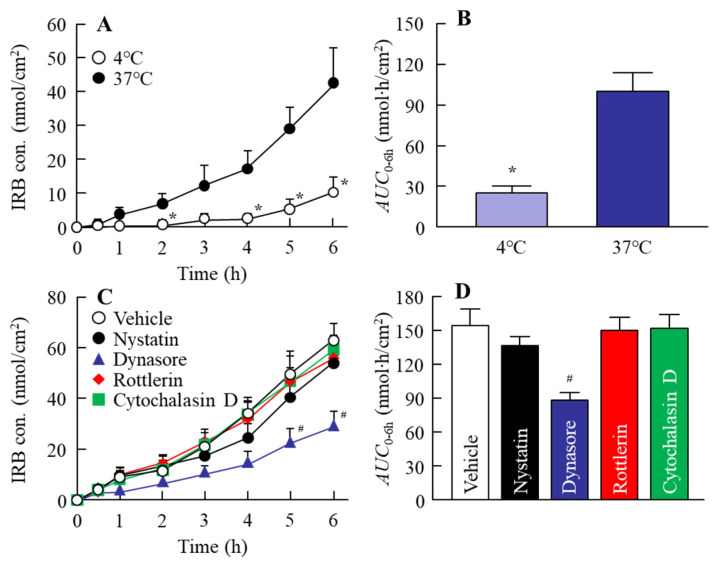
Changes in intestinal penetration of the IRB-NP tablet (Rp.8) through inhibition of endocytosis. (**A**) Penetration profile and (**B**) *AUC*_0–6 h_ of IRB in the IRB-NP tablet under cold (4 °C) and normal (37 °C) conditions. IRB-NP tablet-treated intestine at 4 °C. IRB-NP tablet-treated intestine at 37 °C; (**C**) drug penetration profile and (**D**) *AUC*_0–6 h_ of IRB in the intestines of rats co-treated with an IRB-NP tablet and endocytosis inhibitors. Vehicle, intestine co-treated with 0.5% DMSO, and an IRB-NP tablet. Nystatin, intestine co-treated with nystatin and an IRB-NP tablet. Dynasore, intestine co-treated with dynasore and an IRB-NP tablet. Rottlerin, intestine co-treated with rottlerin and an IRB-NP tablet. Cytochalasin D, intestine co-treated with cytochalasin D and an IRB-NP tablet. *n* = 6–9. * *p* < 0.05 vs. 37 °C for each category. # *p* < 0.05 vs. the vehicle for each category. The dynasore prevented intestinal penetration of the IRB-NP tablet.

**Figure 7 pharmaceutics-14-00387-f007:**
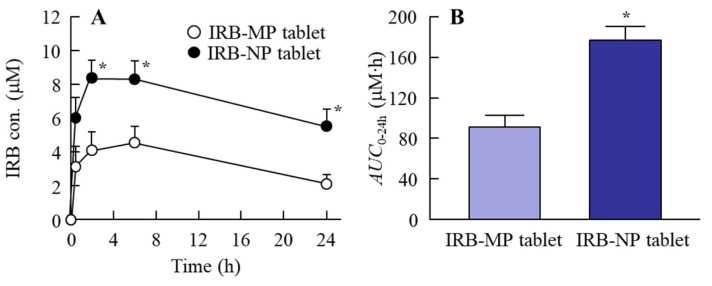
(**A**) The absorption profile and (**B**) *AUC*_0–24 h_ of IRB in the rats orally administered the IRB-MP and IRB-NP tablets (Rp.8). IRB-MP tablet, IRB-MP tablet-administered rats. IRB-NP tablet, IRB-NP tablet-administered rats. *n* = 6–7. * *p* < 0.05 vs. IRB-MP tablet for each category. The intestinal absorption of IRB in the IRB-NP tablet-administered rats was significantly enhanced in comparison with that of the IRB-MP tablet-administered rats.

**Figure 8 pharmaceutics-14-00387-f008:**
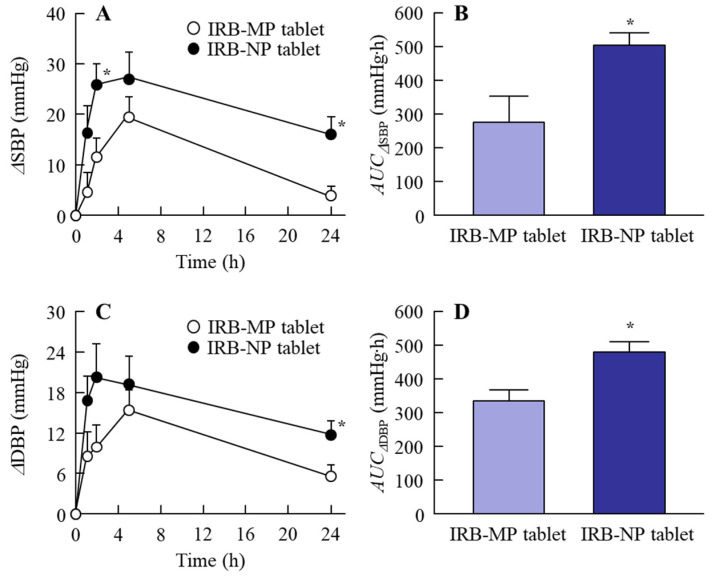
BP and DBP in the SHR-SP rats orally administered the IRB-NP tablet (Rp.8) shown in Table 1. (**A**,**B**) ΔSBP profile (**A**) and *AUC*_ΔSBP_ (**B**) in the SHR-SP rats orally administered the IRB-NP tablet. (**C**,**D**) ΔDBP profile (**C**) and *AUC*_ΔDBP_ (**D**) in the SHR-SP rats orally administered the IRB-NP tablet. *n* = 5–7. * *p* < 0.05 vs. IRB-MP tablet for each category. The BP-reducing effect of the IRB-NP tablet was significantly increased in comparison with the IRB-MP tablet.

**Figure 9 pharmaceutics-14-00387-f009:**
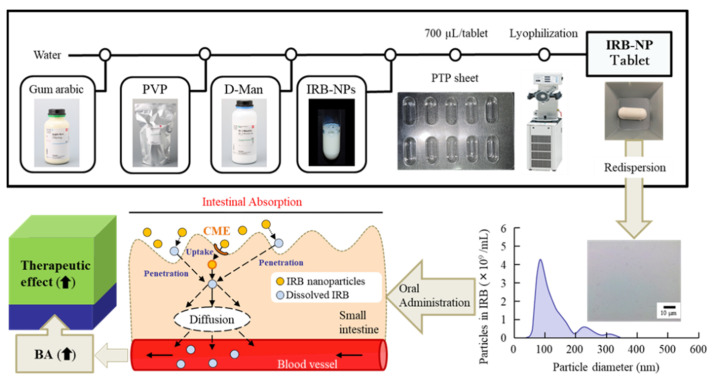
Scheme of the IRB-NP tablet production and intestinal absorption pathways.

**Table 1 pharmaceutics-14-00387-t001:** Compositions of IRB suspensions used to prepare the tablet in this study.

Formulation		Content (*w*/*w*%)						Treatment
		IRB	MC	HPβCD	Mannitol	PVP	Gum Arabic	
Rp.1 (IRB-MP suspensions)		1	0.5	5				―
IRB-MP tablet		1	0.5	5	4	0.4	10	Freeze-drying
Rp.2 (IRB-NP suspensions)		1	0.5	5				Bead mill
IRB-NP tablet	Rp.3	1	0.5	5	4			Bead mill, Freeze-drying
	Rp.4	1	0.5	5		0.4		Bead mill, Freeze-drying
	Rp.5	1	0.5	5			6	Bead mill, Freeze-drying
	Rp.6	1	0.5	5	4		6	Bead mill, Freeze-drying
	Rp.7	1	0.5	5	4	0.4	6	Bead mill, Freeze-drying
	Rp.8	1	0.5	5	4	0.4	10	Bead mill, Freeze-drying
	Rp.9	1	0.5	5	4	0.4	12	Bead mill, Freeze-drying
	Rp.10	1	0.5	5	4	0.4	16	Bead mill, Freeze-drying

**Table 2 pharmaceutics-14-00387-t002:** Characteristics of the redispersed IRB tablet.

Formulation		Particle Size		Solubility	Viscosity	Zeta Potentials
		Mean	Range	µm	Pa∙s	mV
Rp.1 (IRB-MP suspensions)		4.89 µm	0.51–48 µm	145 ± 5.6	2.7 ± 0.3 *	−45.8 ± 1.0
IRB-MP tablet		5.61 µm	0.50–57 µm	142 ± 5.7	12.5 ± 0.9	−44.1 ± 2.5
Rp.2 (IRB-NP suspensions)		87 nm *	50–202 nm	281 ± 4.9 *	2.9 ± 0.2 *	−45.8 ± 1.0
IRB-NP tablet	Rp.3	153 nm *	58–380 nm	338 ± 6.8 *	1.3 ± 0.2 *	−45.2 ± 0.6
	Rp.4	122 nm *	53–392 nm	337 ± 6.5 *	1.1 ± 0.1 *	−44.3 ± 0.7
	Rp.5	143 nm *	55–426 nm	327 ± 7.1 *	5.4 ± 0.6 *	−44.2 ± 0.6
	Rp.6	147 nm *	55–353 nm	338 ± 6.2 *	7.2 ± 0.6 *	−43.8 ± 1.2
	Rp.7	112 nm *	59–333 nm	336 ± 5.9 *	8.3 ± 0.7 *	−44.4 ± 0.5
	Rp.8	118 nm *	55–345 nm	338 ± 5.1 *	12.0 ± 0.8	−40.6 ± 0.6 *
	Rp.9	119 nm *	53–346 nm	340 ± 6.3 *	18.9 ± 1.2 *	−39.7 ± 0.5 *
	Rp.10	117 nm *	52–345 nm	325 ± 6.5 *	36.7 ± 2.3 *	−38.4 ± 0.9 *

The formulations shown in Table 1 were used in this study, and the measurement were performed at 22 °C. *n* = 7–10. * *p* < 0.05 vs. IRB-MP tablet for each category.

**Table 3 pharmaceutics-14-00387-t003:** Particle sizes of IRB in the redispersions of 1-month-stored IRB-NP tablets.

Formulation	IRB-MP Tablet	Rp.2	Rp.7	Rp.8	Rp.9	Rp.10
Mean particle size (range)	5.11 µm	136 nm	131 nm	138 nm	139 nm	137 nm
	(0.61–57 µm)	(57–228 nm)	(65–359 nm)	(61–348 nm)	(59–341 nm)	(60–338 nm)

The formulations shown in Table 1 were used in this study. *n* = 5.

## Data Availability

Not applicable.

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
