# Peer review of "Oral Formulation Based on Irbesartan Nanocrystals Improve Drug Solubility, Absorbability, and Efficacy"

_pharmaceutics, 2022, doi:10.3390/pharmaceutics14020387_

Round 1

Reviewer 1 Report

The authors have made good effort on the research “Oral Formulation Based on Irbesartan Nanocrystals Increases Drug Solubility, Absorbability, and Efficacy”

  1. The heading seems to be incorrect and to be revised.
  2. The authors not mentioned anything about tablet formulation the process weather molding or punching etc.
  3. The authors mentioned Characterization of the IRB-NP Tablets in heading. But no characterization was done on tablets. All the test was related to nanoparticle solution.
  4. All the standard evaluation tests for tablets should be included like dissolution disintegration friability etc.
  5. Mention the rationale of inhibiting energy-dependent endocytosis. What is the correlation with real time studies? The use of pharmacological inhibitors also should be justified.

Author Response

We carefully revised our manuscript according to the suggestions of the reviewer 1, and details are as follows.

Q1. The heading seems to be incorrect and to be revised.

A1. Thank you very much for pointing this out. In order to respond to the reviewer’s comment, we revised the title to “Oral Formulation Based on Irbesartan Nanocrystals Improve Drug Solubility, Absorbability, and Efficacy” from “Design of Oral Formulation Based on Irbesartan Nanocrystals Increases Drug Solubility, Absorbability, and Efficacy”.

Q2. The authors not mentioned anything about tablet formulation the process weather molding or punching etc.

A2. The reviewer’s comment is correct. In this study, we prepared the molded tablets by using the freeze-drying treatments (drying approach). In order to respond to the reviewer’s comment, we added the contents in the Materials and Methods (line 96-112).

Q3. The authors mentioned “Characterization of the IRB-NP Tablets” in heading. But no characterization was done on tablets. All the test was related to nanoparticle solution.

A3. The reviewer’s comments are very important. The XRD, TG-DTA, dissolution, disintegration and friability show the evaluation of the characterization in solid IRB (tablets). Therefore, we revised the heading to “Characterization of the IRB-NP Tablets and It’s Redispersion” from “Characterization of the IRB-NP Tablets” (line 118).

Q4. All the standard evaluation tests for tablets should be included like dissolution disintegration friability etc.

A4. The reviewer’s comment is correct. We measured the dissolution, disintegration and friability of IRB tablet according to the eighteenth edition of the Japanese Pharmacopoeia (JP) https://www.mhlw.go.jp/content/11120000/000788359.pdf. (Ref. 30) The dissolution ratio and disintegration time were 99.9±0.4%, 6±1 sec, respectively. The data of friability was shown in the Figure 3C. In order to respond to the reviewer’s comment, we added these date, and mentioned these methods (line 143-149, 235-238, Reference 30).

Q5. Mention the rationale of inhibiting energy-dependent endocytosis. What is the correlation with real time studies? The use of pharmacological inhibitors also should be justified.

A5. Thank you very much for pointing this out. Previous studies showed that the function of energy-dependent endocytosis in the small intestine was universally inhibited via incubation under cold conditions (4 °C) (Ref. 31). Moreover, it was reported that the nystatin, dynasore, rottlerin, and cytochalasin D were used as pharmacological inhibitors of energy-dependent endocytosis, and the caveolae-dependent endocytosis (CavME), clathrin-dependent endocytosis (CME), macropinocytosis (MP), and phagocytosis were inhibited by the concentration of nystatin (54 μM) (Ref. 32), dynasore (40 μM) (Ref. 33), rottlerin (2 μM) (Ref. 34), and cytochalasin D (10 μM) (Ref. 32), respectively. In addition, our previous studies showed that the treatment by the high concentration and extended treatment time of pharmacological inhibitors caused the stimulation in the intestine, and the intestinal uptake of irbesartan nanocrystalline suspensions via energy-dependent endocytosis were prevented by the protocol used this study (cold conditions and pharmacological inhibitors) (Ref. 19). From these findings, we selected the protocol by cold condition and pharmacological inhibitors to inhibit the energy-dependent endocytosis. In order to respond to the reviewer’s comment, we showed these contents and reference (line 161-166, Reference 19, 31-34).

Thank you for great comments.

Reviewer 2 Report

Manuscript ID: pharmaceutics-1569048

Title: Design of Oral Formulation Based on Irbesartan Nanocrystals Increases Drug Solubility, Absorbability, and Efficacy

The irbesartan-nanoparticle (IRB-NP) tablet with hardness (42.6 N) was developed by combining various additives such as methylcellu- 20 lose, 2-hydroxypropyl-β-cyclodextrin HPβCD, D-mannitol, polyvinylpyrrolidone, and gum Arabic, followed by bead-milling and freeze-drying treatments. The mean particle size in the redispersions of the IRB-NP tablet was approximately 118 nm. The solubility and intestinal absorption of IRB in the IRB-NP tablet were significantly enhanced, and both solubility and clathrin-dependent endocytosis helped improve the low bioavailability of IRB.  In addition, the blood pressure reducing effect of the IRB-NP tablet was significantly higher.

Comments and Questions:

  1. In addition to the PXRD patterns, and TGA and DTA scans of the formulated tablets in Figure 5, should the PXRD patterns and TGA and DTA scans of IRB-MP and IRB-NP be given as well? What are the polymorphs and the degree of crystallinity of IRB-MP and IRB-NP (ie. IRB before and after milling)?
  2. The tableting procedure is unclear and a full elaboration of Figure 9 is required. What is a PTP sheet?  How was the shape/dimensions of the tablet obtained?
  3. Are bead milling and freeze drying lab-scale techniques? Why not homogenization, jet milling, spray drying and supercritical fluid?
  4. Why the conventional pharmaceutical solid dispersion technology was not considered here for the manufacturing of amorphous IRB, or IRB-NP, or IRB solid solution, or IRB-molecular solution, or IRB-eutectic mixture instead?
  5. Are in vitro dissolution rate study and dissolution model also essential?
  6. Should RH% and temperature stability tests be done on the IRB nanoparticles in the tablets?
  7. Why the values of w/w% for the ingredients of the formulation in Table 1 did not add up to 100%?

Author Response

We carefully revised our manuscript according to the suggestions of the reviewer 2, and details are as follows.

Q1. In addition to the PXRD patterns, and TGA and DTA scans of the formulated tablets in Figure 5, should the PXRD patterns and TGA and DTA scans of IRB-MP and IRB-NP be given as well? What are the polymorphs and the degree of crystallinity of IRB-MP and IRB-NP (ie. IRB before and after milling)?

A1. The reviewer’s comment is correct. We previously reported the XRD and TG-DTA data of IRB with (NP) or without (MP) bead mill treatment (Ref. 19). The results showed that the peak patterns of XRD were different between the IRB-MP and IRB-NP suspensions. The melting point of IRB-MP shifted to 177℃ from 182℃ upon the addition of additives, although no difference was observed between the IRB-MP and IRB-NP suspensions. Therefore, we have omitted the results for IRB-MP and IRB-NP suspensions in this study. Thank you very much for pointing this out.

Q2. The tableting procedure is unclear and a full elaboration of Figure 9 is required. What is a PTP sheet? How was the shape/dimensions of the tablet obtained?

A2. The reviewer’s comments are very important. The PTP shows the press through pack (PTP) sheet. The shape/dimensions of the tablet were analyzed by the image analyzing software Image J. In order to respond to reviewer’s comment, we defined the PTP sheet, and added the information in the Material and Methods (line 96, 110-112).

Q3. Are bead milling and freeze drying lab-scale techniques? Why not homogenization, jet milling, spray drying and supercritical fluid?

A3. Thank you very much for pointing this out. High crushing force is required to prepare the solid nanocrystals with particles of 100 nm or less. Therefore, in this study, the bead mill method was applied, since the crushing force by bead mill method tend to be higher in comparison with other milling technique, such as jet milling, spray drying and supercritical fluid. In fact, the particle size of IRB was approximately 200-600 nm after crushing with a microfluidizer (supercritical fluid) using the same additive formulation. Thank you for pointing out this.

Q4. Why the conventional pharmaceutical solid dispersion technology was not considered here for the manufacturing of amorphous IRB, or IRB-NP, or IRB solid solution, or IRB-molecular solution, or IRB-eutectic mixture instead?

A4. In this study, we attempted to design a new method of drug nanoparticulation and its application. For this purpose, we prepared molded tablet by using these additives, bead milling and freeze-drying methods, and evaluated their absorption. Thank you very much for pointing this out.

Q5. Are in vitro dissolution rate study and dissolution model also essential?

A5. The reviewer’s comment is correct. We measured the dissolution according to the Eighteenth edition of the Japanese Pharmacopoeia (JP) https://www.mhlw.go.jp/content/11120000/000788359.pdf. (Ref. 30). The dissolution ratio was 99.9±0.4%. In order to respond to the reviewer’s comment, we added the data in the Discussion (line 143-149, 235-238, Reference 30).

Q6. Should RH% and temperature stability tests be done on the IRB nanoparticles in the tablets?

A6. Thank you for pointing out this. We don’t have data for the RH% and temperature stability tests. On the other hand, the dissolve of IRB-NP tablet was rapidly (6±1 sec), and it was though that the molded tablet is sensitive to humidity and disintegrate quickly. As an alternative to these stability tests, we have added data for stability in 3-month-stored IRB-NP tablets (supplemental data 1), and showed the importance of RH% and temperature stability tests in the Discussion (line 382-390, Supplemental data 1).

Q7. Why the values of w/w% for the ingredients of the formulation in Table 1 did not add up to 100%?

A7. The reviewer’s comment is correct. Table 1 show the compositions of IRB suspensions used to prepare the tablet in this study. We corrected the Table 1 title. Thank you for pointing out this (Table 1).

Thank you for great comments.

Reviewer 3 Report

This manuscript is an interesting work about the use of nanosuspensions to improve the bioavailability of BCS Class II drugs, but I consider that is necessary to improve it to be considered for publication in pharmaceutics 

2.1 Chemical and animals:

Change “two-hydroxypropyl-β-cyclodextrin” by “2- hydroxypropyl-β-cyclodextrin (HPβCD)” and include the DS (degree of substitution) of the cyclodextrin.

 PVP and methylcellulose molecular weight must be included in the text.

The preparation of the tablets was not described in the methodology section. What type of tableting machine was used to obtain tablets? what were the conditions of the tableting process?

Could you please include the permeation area or the dimensions of the intestinal tissue, and the volume of the donor and receptor compartment used in the ex vivo permeability studies?

Related to the oral administration of the animals, why were the tablets redispersed? It was not possible to administrate the solid form (or a portion of the tablet) using an oral sonde? The authors must introduce a commentary about the influence of the in vivo tablet disaggregation on the rate of absorption. Was the disaggregation time of the tablets determined?

Why was IRB-MP not studied in the Transintestinal Penetration experiments?
Page 11 line 367. The authors write “the intestinal absorption of IRB via the methacrylate cell set on the removed rat skin (Figure 6)”  I guess it's a mistake and rat skin was included instead of rat gut. Correct it

The stability studies of 1-month-stored IRB-NP tablets are not enough to characterize the ability of tablets to stabilize the Nanosuspensions. Authors must complete this study prolonging the storage time and monitoring changes in the main properties of tables including redispersion capacity,  disaggregation time, friability, hardness, or microstructural changes (using x-ray, DTA..) 

Author Response

We carefully revised our manuscript according to the suggestions of the reviewer 3, and details are as follows.

Q1. Change “two-hydroxypropyl-β-cyclodextrin” by “2-hydroxypropyl-β-cyclodextrin (HPβCD)” and include the DS (degree of substitution) of the cyclodextrin.

A1. The reviewer’s comments are very important. We revised to “2-hydroxypropyl-β-cyclodextrin (HPβCD)”. In addition, we added the average molar substitution (0.6) and average molecular weight (1380) of HPβCD (line 74-76).

Q2. PVP and methylcellulose molecular weight must be included in the text.

A2. The reviewer’s comment is correct. The average molecular weight of PVP and methylcellulose are 40,000 and 25,000, respectively. In order to respond to the reviewer’s comment, we added the information (line 79-82).

Q3. The preparation of the tablets was not described in the methodology section. What type of tableting machine was used to obtain tablets? what were the conditions of the tableting process?

A3. Thank you very much for pointing this out. In this study, we prepared the molded tablets by using the freeze-drying treatments (drying approach), and the FREEZE DRYER FD-1000 (TOKYO RIKAKIKAI Co., Ltd., Tokyo, Japan) was used (no punching or other processes are used for the preparation of tablet based on irbesartan nanocrystals). In order to respond to the reviewer’s comment, we added the contents in the Materials and Methods (line 96-109).

Q4. Could you please include the permeation area or the dimensions of the intestinal tissue, and the volume of the donor and receptor compartment used in the ex vivo permeability studies?

A4. Thank you for pointing out this. The permeation area and dimensions of the intestinal tissue are 0.3485 cm2, 0.1 cm (average of 5 rats), respectively. Moreover, the volume of the donor and receptor compartment is 3 mL. In order to respond to the reviewer’s comment, we added the information in the Materials and Methods (line 153-157).

Q5. Related to the oral administration of the animals, why were the tablets redispersed? It was not possible to administrate the solid form (or a portion of the tablet) using an oral sonde? The authors must introduce a commentary about the influence of the in vivo tablet disaggregation on the rate of absorption. Was the disaggregation time of the tablets determined?

A5. The reviewer’s comments are very important. The dissolve of IRB-NP tablet is rapidly (6±1 sec), and is not expected to affect the absorption rate. From these results, we administrated the redispersed IRB-NP tablet. In order to respond to the reviewer’s comment, we added the data for disaggregation time of the tablets (line 235-238).

Q6. Why was IRB-MP not studied in the Transintestinal Penetration experiments?

Page 11 line 367. The authors write “the intestinal absorption of IRB via the methacrylate cell set on the removed rat skin (Figure 6)” I guess it's a mistake and rat skin was included instead of rat gut. Correct it.

A6. The reviewer’s comment is correct. We corrected to “the intestinal absorption of IRB via the methacrylate cell set on the removed rat gut (Figure 6)” from “the intestinal absorption of IRB via the methacrylate cell set on the removed rat skin (Figure 6)”. In addition, in order to respond to the reviewer’s comment, we added the data (AUC0-6h, 36±9 nmol∙h/cm2) of in vitro transintestinal penetration in IRB-MP tablet (line 289-291, 397).

Q7. The stability studies of 1-month-stored IRB-NP tablets are not enough to characterize the ability of tablets to stabilize the Nanosuspensions. Authors must complete this study prolonging the storage time and monitoring changes in the main properties of tables including redispersion capacity, disaggregation time, friability, hardness, or microstructural changes (using x-ray, DTA.)

A7. Thank you very much for pointing this out. We added the data for redispersion capacity (supplemental data 1A), disaggregation time (6±1 sec, n=5), friability (5.1±0.5%, n=5), hardness (42±6 N, n=5) in 3-month-stored IRB-NP tablets (Rp.8 formulation). In addition, we measured the XRD pattern of 3-month-stored IRB-NP tablets (supplemental data 1B). On the other hand, it is important to evaluate the change in physical properties when stored for a long period of time. In order to respond to the reviewer’s comment, we added these data and the importance of measuring the physical properties after long-term storage in the Discussion. Thank you for pointing out this. (line 382-390, Supplemental data 1).

Thank you for great comments.

Round 2

Reviewer 1 Report

The responses to the comments are adequate

Reviewer 2 Report

Accept in present form.

Reviewer 3 Report

thank the authors for having taken into account my considerations in the review. I consider that the article can be published in Pharmaceutics